Synthesis and anti-tubercular activity of 3-substituted benzo[b]thiophene-1,1-dioxides

Chandrasekera N. Susantha
Bailey Mai A.
Files Megan
Alling Torey
Florio Stephanie K.
Ollinger Juliane
Odingo Joshua O.
Parish Tanya tanya.parish@idri.org
TB Discovery Research, Infectious Disease Research Institute , Seattle, WA , USA
Tulkens Paul
Electronic publication date: 2014 Oct 7
Publication date: 2014
Volume: 2
Electronic Location ID: e612
Received 2014 Aug 2; Accepted 2014 Sep 16
Copyright: © 2014 Chandrasekera et al.
Copyright year: 2014
Copyright holder: Chandrasekera et al.
License: This is an open access article distributed under the terms of the Creative Commons Attribution License, which permits unrestricted use, distribution, reproduction and adaptation in any medium and for any purpose provided that it is properly attributed. For attribution, the original author(s), title, publication source (PeerJ) and either DOI or URL of the article must be cited.
License URL: https://creativecommons.org/licenses/by/4.0/

Keywords: Tuberculosis, Antimicrobial, Benzothiophene dioxide, Drug discovery, Mycobacterium tuberculosis, High throughput screening, Antibacterial

Funding: Bill & Melinda Gates Foundation OPP1024038 This research was funded by grant OPP1024038 from the Bill & Melinda Gates Foundation. The funders had no role in study design, data collection and analysis, decision to publish, or preparation of the manuscript.

==============================
We demonstrated that the 3-substituted benzothiophene-1,1-dioxide class of compounds are effective inhibitors of Mycobacterium tuberculosis growth under aerobic conditions. We examined substitution at the C-3 position of the benzothiophene-1,1-dioxide series systematically to delineate structure–activity relationships influencing potency and cytotoxicity. Compounds were tested for inhibitory activity against virulent M. tuberculosis and eukaryotic cells. The tetrazole substituent was most potent, with a minimum inhibitory concentration (MIC) of 2.6 µM. However, cytotoxicity was noted with even more potency (Vero cell TC50 = 0.1 µM). Oxadiazoles had good anti-tubercular activity (MICs of 3–8 µM), but imidazoles, thiadiazoles and thiazoles had little activity. Cytotoxicity did not track with anti-tubercular activity, suggesting different targets or mode of action between bacterial and eukaryotic cells. However, we were unable to derive analogs without cytotoxicity; all compounds synthesized were cytotoxic (TC50 of 0.1–5 µM). We conclude that cytotoxicity is a liability in this series precluding it from further development. However, the series has potent anti-tubercular activity and future efforts towards identifying the mode of action could result in the identification of novel drug targets.

Introduction

Tuberculosis (TB), which is caused by Mycobacterium tuberculosis, is the second leading cause of death from an infectious disease and is a major global health problem. In 2010, according to the World Health Organization (WHO) 8.8 million new cases and 1.4 million deaths from the disease were reported (WHO, 2011). In addition, one third of the world population has latent TB, 10% of whom are expected to develop active TB at some point in their lives. Currently the recommended first-line TB treatment regimens require a minimum of 6 months of multidrug therapy, resulting in challenges with patient adherence. The result of inadequate therapy and poor compliance has contributed to a rise in the emergence of multidrug resistant (MDR), resistant to isoniazid and rifampicin, and extensively drug-resistant (XDR) strains, resistant to a fluoroquinolone and at least one injectable drug, of M. tuberculosis (WHO, 2011). Consequently, there is an urgent need for the development of novel anti-TB drugs that are effective against both drug sensitive and resistant M. tuberculosis (Ginsberg, 2010).

The benzo[b]thiophene-1,1-dioxide (BTD) series was reported to have activity against M. tuberculosis in a phenotypic assay (Ananthan et al., 2009). Fourteen compounds were tested from this series; five of these, all of which had heteroarylthio groups, had some inhibitory activity against M. tuberculosis. As a part of our ongoing TB drug discovery program, we were interested in exploring the potential of the BTD series to be developed as a lead series for TB treatment. We conducted an exploratory chemistry study and evaluated the series for their activity against M. tuberculosis as well as cytotoxicity for eukaryotic cells.

Materials and Methods

Determination of minimum inhibitory concentration (MIC)

We used M. tuberculosis H37Rv (London Pride), a laboratory-passaged derivative of H37Rv (ATCC 25618), which has been sequenced, as described in Ioerger et al. (2010). MICs were run as described (Ollinger et al., 2013); briefly MICs were determined against M. tuberculosis grown in Middlebrook 7H9 medium containing 10% OADC (oleic acid, albumin, dextrose, catalase) supplement (Becton Dickinson) and 0.05% w/v Tween 80 (7H9-Tw-OADC) under aerobic conditions. Compounds were prepared as 10-point two-fold serial dilutions in DMSO with a starting concentration of 20 µM (lowest compound concentration 40 nM). The final concentration of DMSO in the assay was 2%. Bacterial growth was measured by OD590 after 5 days of incubation at 37 °C and % growth measured. Growth inhibition curves were plotted and fitted using the Gompertz model. The MIC was defined as the minimum concentration required for >99% growth inhibition.

Vero cytotoxicity assay

CellTiter-Glo® Luminescent Cell Viability Assay (Promega) was used to measure ATP as a indicator of cell viability. The Vero cell line (ATCC CCL81) was grown in Dulbecco’s Modified Eagle Medium (DMEM), High Glucose, GlutaMAX™ (Invitrogen), 10% FBS (Fetal Bovine Serum), and 1x of Penicillin-Streptomycin Solution (100 units/mL of penicillin, 100 µ g/mL of streptomycin). Compounds were solubilized in DMSO (dimethyl sulfoxide) and assayed using a 10-point three-fold serial dilution starting at the highest concentration of 50 µ M. CellTiter-Glo® Reagent (Promega) was added to 96-well plates after 2 days of incubation at 37 °C, 5% CO2. Relative luminescent units (RLU) were measured using Perkin Elmer Wallac 1420 Victor2 plate reader. Inhibition curves were fitted using the Levenberg–Marquardt algorithm. Toxic concentration (TC50) was defined as the concentration of compound that gave 50% inhibition of growth. Selectivity index was calculated as MIC/TC50. For published data (Ananthan et al., 2009), SI was calculated as IC90/TC50.

Analysis of compounds

1H and NMR spectral data were recorded in CDCl3 or Acetone-d6 on a 300 MHz Bruker NMR spectrometer. Column chromatography was conducted on a Revelaris flash chromatography system. Reactions were monitored using thin-layer chromatography (TLC) on silica gel plates. HPLC analysis was conducted on an Agilent 1100 series LC system (Agilent ChemStation Rev.A.10.02; Phenomenex-Luna-C18, 4.8 mm × 150 mm, 5 µm, 1.0 mL/min, UV 254 nm, room temperature) with MeCN/H2O (0.05% TFA or HCOOH buffer) gradient elution. HPLC-MS was performed on a Gilson 321 HPLC with detection performed by a Gilson 170 DAD and a Finnigan AQA mass spectrometer operating in electrospray ionisation mode using a Phenomenex Gemini C18 150 × 4.6 mm column. Compounds 3a, b, c, s, t and u were purchased from ChemBridge Corporation.

Synthesis of 3-bromobenzo[b]thiophene 1,1-dioxide (2)

To a solution of 1 1.62 g (7.6 mmol) in 25.0 mL of acetic acid was added 30% aqueous hydrogen peroxide and the mixture was heated for 1 h at 100  °C. The mixture was poured into ice cold water and let stand overnight. The resulting solid was filtered and dried to yield 2 (1.65 g, 89%). 1H NMR (300 MHz, CDCl3): δ 6.98 (s, 1H), 7.58–7.72 (m, 4H). LCMS–ESI (M+ H)+: 214.1.

General procedure for the synthesis of 3-substituted benzo[b]thiophene-1,1-dioxides

To a solution of 200 mg (0.82 mmol) of 2 in 5 mL of dimethyl formamide was added 2.0 mmol of the thiol reagent followed by 0.5 mL of triethylamine. The reaction was stirred overnight and washed with 20 mL of deionized water and extracted with 50 mL of ethyl acetate. The organic layer was dried with anhydrous sodium sulfate, filtered and concentrated in vacuo. The resulting residue was purified by reveleris flash chromatography system to yield the aryl/heteroaryl thio benzo[b]thiophene 1,1-dioxides.

3-((5-(4-methoxyphenyl)-1,3,4-oxadiazol-2-yl)thio)benzo[b]thiophene 1,1-dioxide (3d)

Yield 3d: (95 mg, 31%). 1H NMR (300 MHz, CDCl3): 3.9 (3H, OCH3. s); 7.0–8.0 (m, 9H). LCMS–ESI (M+ H)+: 373.0.

3-((5-(4-chlorophenyl)-1,3,4-oxadiazol-2-yl)thio)benzo[b]thiophene 1,1-dioxide (3e)

Yield 3e: (122 mg, 39%). 1H NMR (300 MHz, Methanol-d4): 7.5–8.1 (m, 8H). LCMS–ESI (M+ H)+: 377.0.

3-(thiazol-2-ylthio)benzo[b]thiophene 1,1-dioxide (3f)

Yield 3f: (87 mg, 31%). 1H NMR (300 MHz, CDCl3): 6.6 (1H, s); 7.6–8.1 (m, 6H). LCMS–ESI (M+ H)+: 282.0.

3-((4-phenylthiazol-2-yl)thio)benzo[b]thiophene 1,1-dioxide (3g)

Yield 3g: (25 mg, 9%). 1H NMR (300 MHz, CDCl3): 7.7–8.0 (m, 10H). LCMS–ESI (M+ H)+: 358.0.

3-(benzo[d]thiazol-2-ylthio)benzo[b]thiophene 1,1-dioxide (3h)

Yield 3h: (65 mg, 32%). 1H NMR (300 MHz, CDCl3): 7.3–8.1 (m, 9H). LCMS–ESI (M+ H)+: 332.0.

3-((5-chloro-3a,7a-dihydrobenzo[d]thiazol-2-yl)thio)benzo[b]thiophene 1,1-dioxide (3i)

Yield 3i: (25 mg, 30%). 1H NMR (300 MHz, CDCl3): 7.4 (1H, s); 7.5–8.1 (m, 7H). LCMS–ESI (2M+H2O)+: 754.9.

3-((6-ethoxybenzo[d]thiazol-2-yl)thio)benzo[b]thiophene 1,1-dioxide (3j)

Yield 3j: (105 mg, 28%). 1H NMR (300 MHz, CDCl3): 1.5 (3H, d); 4.1 (2H, t); 7.1–7.9 (m, 8H). LCMS–ESI (M+ H)+: 376.0.

3-((5-methyl-1,3,4-thiadiazol-2-yl)thio)benzo[b]thiophene 1,1-dioxide (3k)

Yield 3k: (26 mg, 11%). 1H NMR (300 MHz, CDCl3): 2.6 (3H, CH3, s); 7.7–8.0 (m, 9H). LCMS–ESI (M+ H)+: 297.0.

3-((5-amino-1,3,4-thiadiazol-2-yl)thio)benzo[b]thiophene 1,1-dioxide (3l)

Yield 3l: (16 mg, 7%). 1H NMR (300 MHz, CDCl3): 6.8 (1H, s); 7.3 (2H, NH2, s); 7.6–7.8 (m, 4H). Yield 20: (5 mg, 2%). LCMS–ESI (M+ H)+: 298.0.

3-((5-mercapto-1,3,4-thiadiazol-2-yl)amino)benzo[b]thiophene 1,1-dioxide (3m)

Yield 3m: (5 mg, 2%). 1H NMR (300 MHz, CDCl3): 7.1 (2H, s); 7.6–7.8 (m, 4H); 8.8 (1H, SH, s). LCMS–ESI (M+ H)+: 298.0.

3-((1H-benzo[d]imidazol-2-yl)thio)benzo[b]thiophene 1,1-dioxide (3n)

Yield 3n: (100 mg, 39%). 1H NMR (300 MHz, CDCl3): 6.7 (1H, s); 7.3–7.8 (m, 9H). LCMS–ESI (M+ H)+: 315.0.

3-((1-methyl-3a,7a-dihydro-1H-benzo[d]imidazol-2-yl)thio)benzo[b]thiophene 1,1-dioxide (3o)

Yield 3o: (57 mg, 21%). 1H NMR (300 MHz, CDCl3): 3.9 (3H, CH3. s); 6.4 (1H, s); 7.3–7.8 (m, 8H). LCMS–ESI (M+ H)+: 329.0.

3-((5-nitro-3a,7a-dihydro-1H-benzo[d]imidazol-2-yl)thio)benzo[b]thiophene 1,1-dioxide (3p)

Yield 3p: (142 mg, 48%). 1H NMR (300 MHz, CDCl3): 7.5 (1H, s); 7.7–7.9 (m, 4H); 8.2 (2H, d); 8.5 (1H, s). LCMS–ESI (M+ H)+: 360.0.

3-((5-methoxy-3a,7a-dihydro-1H-benzo[d]imidazol-2-yl)thio)benzo[b]thiophene 1,1-dioxide (3q)

Yield 3q: (97 mg, 34%). 1H NMR (300 MHz, Methanol-d4): 3.8 (3H, OCH3. s); 6.6 (1H, s); 7.0–7.7 (m, 7H).

3-((1-methyl-1H-tetrazol-5-yl)thio)benzo[b]thiophene 1,1-dioxide (3r)

Yield 3r: (115 mg, 50%). 1H NMR (300 MHz, CDCl3): 4.3 (3H, 3CH3, s); 6.9 (1H, s); 7.5–8.1 (m, 7H). LCMS–ESI (2M+ H)+: 561.0.

3-(pyridin-2-ylthio)benzo[b]thiophene 1,1-dioxide (3v)

Yield 3v: (120 mg, 53%). 1H NMR (300 MHz, CDCl3): 6.6–8.5 (m, 9H). LCMS–ESI (M+ H)+: 276.0.

3-(pyridin-4-ylthio)benzo[b]thiophene 1,1-dioxide (3w)

Yield 3w: (57 mg, 25%). 1H NMR (300 MHz, CDCl3): 6.6 – 8.5 (m, 9H). LCMS–ESI (M+ H)+: 276.0.

3-(isoquinolin-3-ylthio)benzo[b]thiophene 1,1-dioxide (3x)

Yield 3x: (95 mg, 36%). 1H NMR (300 MHz, CDCl3): 7.3–8.1 (m, 6H).

3-(naphthalen-2-ylthio)benzo[b]thiophene 1,1-dioxide (3y)

Yield 3y: (110 mg, 42%). 1H NMR (300 MHz, CDCl3): 5.8 (1H, s); 7.6–8.1 (m, 11H). LCMS–ESI (M+2Na)+: 671.0.

Results and Discussion

BTD analogs were synthesized as outlined in Fig. 1. The oxidation of commercially available 3-bromothianaphthalene (1) with hydrogen peroxide afforded 3-bromobenzothiophene-1,1-dioxide (2). This in turn was reacted with the corresponding thiols to afford the 3-substituted BTDs. To investigate the biological activity, we conducted a systematic exploration of the aryl/heteroaryl substituents linked via a thioether to the C-3 position of the benzo[b]thiophene-1,1-dioxide compound.

Figure 1 Synthesis of 3-substituted benzo[b]thiophene-1,1-dioxides.

We probed the consequences of having oxazoles and oxadiazoles as substituents at the C-3 position. Compounds were tested for efficacy against a virulent strain of M. tuberculosis in liquid culture under aerobic growth conditions (Ollinger et al., 2013). All compounds had good activity and the minimum inhibitory concentration (MIC) was very similar (3–8 µ M) (Table 1). The change in electronics of the phenyl substituents had no effect on potency of the oxadiazole compounds. The addition of the electron donating groups, methyl (3b), methoxy (3d) or an electron withdrawing Cl-group (3e) to the para position of (3a) resulted in similar MIC values (Table 1). MICs were similar for benzaoxazole 3c and the phenyl linked oxadiazoles (3a, b, d, and e). This confirmed that the series has good anti-tubercular activity. We tested compound activity against eukaryotic cells using the Vero cell line (derived from African green monkey kidney cells). All of the compounds had significant cytotoxicity, with TC50 values <0.3 µ M, suggesting that these compounds are even more effective against eukaryotic cells (Table 1). Of the compounds we tested, two had previously been identified as having anti-tubercular activity (3a and 3i)(Ananthan et al., 2009). In this study 3a appeared to have a selectivity index (SI) of >33. However, in our assay this compound had a SI of 0.03. The compound 3a was reported to have anti-tubercular activity with an IC90 of 1.3 µM (0.45 µg/mL) and a TC50 of 43 µM (calculated from the published data using the equation TC50 = SI × IC90). Ananthan et al. (2009) calculated IC90 in their assay, representing the concentration required to inhibit growth by 90%, but in our experience IC90 and MIC (which we used) are very similar. In our case it had an MIC of 3.1 µM and a cytotoxicity of 0.1 µM. Therefore, the difference in SI is primarily due to the difference in cytotoxicity data.

Table 1 Activity of oxazole and oxadiazole analogs of the BTD series against M. tuberculosis and Vero cell line.

Compound	R-group	MIC (µM)a	TC50 (µM)b	SIc	
3a		3.1 ± 0.07	0.1 ± 0	0.03	
3b		8.2 ± 0.6	0.2 ± 0	0.02	
3c		5.7 ± 2.9	0.2 ± 0.07	0.04	
3d		7.2 ± 0.3	0.3 ± 0.3	0.04	
3e		3.9 ± 1.7	0.3 ± 0.2	0.08	
Notes.

a MIC is the minimum concentration required to inhibit growth of M. tuberculosis completely in liquid culture (Ollinger et al., 2013). MICs of active compounds are the average of two independent experiments ± standard deviation.

b TC50 is the concentration required to inhibit growth of Vero cells by 50%. TC50 is the average of two runs ± standard deviation.

c SI is the selectivity index. Selectivity index is calculated as MIC/TC50.

For comparison, MIC of rifampicin is 0.003 µM and isoniazid is 0.2 µM (Ollinger et al., 2013).

Since we had seen good activity with the compounds, but significant cytotoxicity, we determined whether we could separate the two activities to generate potent, non-toxic compounds. We examined the influence of thiazoles and thiadiazoles on the biological activity and cytotoxicity of these BTD compounds. Anti-tubercular activity was diminished by the replacement of an oxazole with either a thiazole or a thiadiazole; these compounds showed MICs ≥20 µM (Table 2). The only exception was compound 3k which showed good activity (9 µM), where the addition of an electron donating ethoxy group to the benzothiazole compound improved its potency to 5 µM (3i). In contrast, addition of an electron-withdrawing group diminished activity in compound 3j (MIC >20 µM, Table 2). Cytotoxicity was also reduced by 10-100-fold, and although the selectivity index (SI) was also improved the compounds were still more active against eukaryotic cells with SI of <0.2 (Table 2). The benzothiazole compound 3i has previously been reported (Ananthan et al., 2009), but in contrast to our results, it had a SI >150, whereas our data indicate that the SI = 0.5. The compound 3i was reported to have a TB IC90 of <0.3 µM (<0.1 µg/mL) and a TC50 of 45 µM. In our assays it gave an MIC of 20 µM and a cytotoxicity of 1 µM. In this case the difference in SI is due to both the difference in activity and cytotoxicity data.

Table 2 Activity of thiazole and thiadiazole analogs of the BTD series against M. tuberculosis and Vero cell line.

Compound	R-group	MIC (µM)a	TC50 (µM)b	SIc	
3g		20	1	0.05	
3h		20	3 ± 1.2	0.2	
3i		20	1	0.05	
3j		>20	1 ± 0.4	NC	
3k		9.0 ± 4.7	1 ± 0.1	0.1	
3l		>20	1 ± 0.4	NC	
3m		>20	3 ± 1	NC	
3n		20	3 ± 0.7	0.2	
Notes.

a MIC is the minimum concentration required to inhibit growth of M. tuberculosis completely in liquid culture (Ollinger et al., 2013). MICs of active compounds are the average of two independent experiments ±standard deviation.

b TC50 is the concentration required to inhibit growth of Vero cells by 50%. TC50 is the average of two runs ±standard deviation.

c SI is the selectivity index. Selectivity index is calculated as MIC/TC50. For comparison, MIC of rifampicin is 0.003 µM and isoniazid is 0.2 µM (Ollinger et al., 2013).

NC Not calculated

We then investigated the effect of C-3 imidazoles to see if we could improve the SI. Similar to the thiazoles and thiadiazoles, this resulted in diminished activity (MIC >20 µM) (3n–3q) (Table 3). Cytotoxicity was similar to those seen with the thiazole and thiadiazole groups. Methylation of the N-1 of the imidazole (3o) had no effect on activity (3n, 3p and 3q). The tetrazole compound (3r) showed the best activity of all the compounds synthesized (MIC = 2.6 µM), but also had significant cytotoxicity (Table 3).

Table 3 Activity of imidazole and tetrazole analogs of the BTD series against M. tuberculosis and Vero cell line.

Compound	R-group	MIC (µM)a	TC50 (µM)b	SIc	
3n		>20	1	NC	
3o		>20	5	NC	
3p		>20	ND	NC	
3q		>20	0.3 ± 0.07	NC	
3r		2.6	0.1 ± 0	0.004	
Notes.

a MIC is the minimum concentration required to inhibit growth of M. tuberculosis completely in liquid culture (Ollinger et al., 2013). MICs of active compounds are the average of two independent experiments ±standard deviation.

b TC50 is the concentration required to inhibit growth of Vero cells by 50%. TC50 is the average of two runs ±standard deviation.

c SI is the selectivity index. Selectivity index is calculated as MIC/TC50.

NC Not calculated

ND Not determined

Finally, we explored the influence of having six membered heterocycles in the C-3 position. We synthesized compounds with pyrimidyl (3t), pyridyl (3v, 3w), quinolinyl (3u), or isoquinolinyl (3x) groups and a non-heterocyclic compound with a naphthyl group (3y). All these analogs were inactive suggesting that the BTD series requires a five membered heterocyclic substituent at the C-3 position linked via a thioether for its activity against M. tuberculosis (Table 4).

Table 4 Activity of six membered heterocyclic analogs of the BTD series against M. tuberculosis and Vero cell line.

Compound	R-group	MIC (µM)a	TC50 (µM)b	SIc	
3s		>20	10 ± 1	NC	
3t		>20	ND	NC	
3u		>20	ND	NC	
3v		>20	ND	NC	
3w		>20	ND	NC	
3x		>20	ND	NC	
3y		>20	ND	NC	
Notes.

a MIC is the minimum concentration required to inhibit growth of M. tuberculosis completely in liquid culture (Ollinger et al., 2013). MICs of active compounds are the average of two independent experiments ±standard deviation.

b TC50 is the concentration required to inhibit growth of Vero cells by 50%. TC50 is the average of two runs ±standard deviation.

c SI is the selectivity index. Selectivity index is calculated as MIC/TC50.

NC Not calculated

ND Not determined

Conclusions

We conducted a systematic exploration of the aryl/heteroaryl thioether substituents at the C-3 position of the benzo[b]thiophene-1,1-dioxide compound series for its inhibitory activity against M. tuberculosis. The series exhibited encouraging activity with some MIC values <10 µM. The tetrazole, oxazole and the oxadiazoles were the most potent compounds tested, whereas compounds bearing six-membered aromatic substituents at the C-3 position were inactive. However, the BTD series was also active against eukaryotic cells showing significant toxicity against the Vero cell line; in fact cytotoxicity was more pronounced than the anti-mycobacterial activity. Our data are in contrast to that previously reported in which cytotoxicity was not observed in selected members of the series (Ananthan et al., 2009). Differences in cytotoxicity could be due to the exact assay method and the cell line used; in this case we used the same Vero cell line. However, the assays conditions were different; we used passaged cells which were actively replicating as opposed to cells recovered directly from frozen. Since the majority of cytotoxicity is manifested during cell division, this may account for our increased sensitivity. In any case, we found that the series as a whole was cytotoxic. We were unable to reduce cytotoxicity in this series, even after significant modifications of the third position substituent. On this basis we concluded that the series lacks further potential for drug development. However, the target of the series may still be of interest, since one might find alternative scaffolds with specificity. Thus, in the future, we are interested in finding the target of these compounds.

Supplemental Information

Supplemental Information 1 Supplementary material

General methods

Click here for additional data file.

We thank Alfredo Blakeley, David Roberts and Yulia Ovechkina for technical assistance.

Additional Information and Declarations

Competing Interests

Author Contributions

Tanya Parish is an Academic Editor for PeerJ. The authors are employees at Infectious Disease Research Institute (IDRI).

N. Susantha Chandrasekera conceived and designed the experiments, performed the experiments, analyzed the data, wrote the paper, prepared figures and/or tables, reviewed drafts of the paper.

Mai A. Bailey and Torey Alling conceived and designed the experiments, performed the experiments, reviewed drafts of the paper.

Megan Files performed the experiments, reviewed drafts of the paper.

Stephanie K. Florio performed the experiments, analyzed the data, reviewed drafts of the paper.

Juliane Ollinger conceived and designed the experiments, performed the experiments, analyzed the data, reviewed drafts of the paper.

Joshua O. Odingo conceived and designed the experiments, analyzed the data, wrote the paper, prepared figures and/or tables, reviewed drafts of the paper.

Tanya Parish conceived and designed the experiments, analyzed the data, wrote the paper, reviewed drafts of the paper.

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
