# Peer review of "Synthesis and anti-tubercular activity of 3-substituted benzo[b]thiophene-1,1-dioxides"

_PeerJ, doi:10.7717/peerj.612_

## Round 0.1 · original submission · Minor Revisions

While one reviewer had no remark, the suggestions and criticisms of the other reviewer suggest that more than minor revision is needed. Please, pay attention to these remarks and explain, point by point, how you have dealt with. A version with all changes limelighted is, to me, also essential.

·

Basic reporting

The manuscript “Synthesis and anti-tubercular activity of 3-substituted benzo[b]thiophene-1,1-dioxides by Chandrasekera S et al,” describes the in vitro screening and cytotoxicity of BTD series against MTB.

Experimental design

The anti TB screening and culture conditions to be explained little more

Validity of the findings

Since authors themselves have concluded that these NCEs are primarily ineffective against MTB, thus it does not leave any space to comment upon the molecules. Although their prospective modifications could be potential as anti-TB compounds.

Additional comments

1. Would be better to know that these compounds do not have more than one enatiomer. This may affect the cytotoxicity issue.
2.The studies are very preliminary but in context of lead optimization it may be useful for the scientific community to design further.
3. The drafting of manuscript is poor and need to be re-written before the final acceptance.
4. Compounds 3a to 3e have low MIC. I’m wondering to know, whether they have been tried lower than 3.12 uM. It would be preferable to provide SI index in the same table.
Recommendations: Manuscript can be accepted after minor revisions.

·

Basic reporting

Well drafted article. Interesting results and can be considered for publication.

Experimental design

Appropriate

Validity of the findings

Good

Additional comments

Article is well drafted. Interesting results and can be considered for publication as it is.

---

## Round 0.2 · accepted · Accept

The present version has been satisfactorily improved oever the first one.